# Voxel-based 3D Detection and Reconstruction of Multiple Objects from a Single Image

**Feng Liu**  **Xiaoming Liu**
Department of Computer Science and Engineering
Michigan State University, East Lansing MI 48824
{liufeng6, liuxm}@msu.edu

## Abstract

Inferring 3D locations and shapes of multiple objects from a single 2D image is a long-standing objective of computer vision. Most of the existing works either predict one of these 3D properties or focus on solving both for a single object. One fundamental challenge lies in how to learn an effective representation of the image that is well-suited for 3D detection and reconstruction. In this work, we propose to learn a regular grid of 3D voxel features from the input image which is aligned with 3D scene space via a 3D feature lifting operator. Based on the 3D voxel features, our novel CenterNet-3D detection head formulates the 3D detection as keypoint detection in the 3D space. Moreover, we devise an efficient coarse-to-fine reconstruction module, including coarse-level voxelization and a novel local PCA-SDF shape representation, which enables fine detail reconstruction and one order of magnitude faster inference than prior methods. With complementary supervision from both 3D detection and reconstruction, one enables the 3D voxel features to be geometry and context preserving, benefiting both tasks. The effectiveness of our approach is demonstrated through 3D detection and reconstruction in single object and multiple object scenarios. Code is available at http://cvlab.cse.msu.edu/project-mdr.html.

## 1 Introduction

As a fundamental computer vision task, instance-level 3D scene understanding from a single image has drawn substantial attention from researchers due to its importance in applications such as robotics [1, 2], AR/VR [3] and autonomous driving [4–6]. The important 3D properties include 3D bounding box (pose, size, location) and 3D shape of object instances. In this work, we aim to design a framework to infer all these 3D properties of multiple objects from a single 2D image.

In recent years, various monocular methods are proposed to predict either 3D boxes [7–12] or 3D shapes [13–17]. However, only a few studies [18–23] consider both 3D detection and reconstruction for a total 3D scene understanding. The complexity of real-world scenarios and diverse category variations make it challenging to fully reconstruct the scene context (both semantics and geometry) at the instance level from a single image. Moreover, those methods primarily assign 3D semantic labels to pixels. Yet, such a 2D representation with depth ambiguity is insufficient for 3D geometry and context reasoning. It is thus crucial to develop an effective representation of the image that is relevant to 3D geometry and spatial information for performing accurate 3D detection and reconstruction.

In light of this, attempts like grid-based representation have been made for tasks such as rendering [24], detection [25, 26], or reconstruction [27]. OFT [25] proposes to sample and transform image features into a BEV grid representation, which enables holistic reasoning of the 3D scene configuration. CaDDN [26] extends the BEV grid representation with a categorical depth prior, leading to higher 3D detection accuracies. DeepVoxels [24] and UCLID-Net [27] build voxel features by back-projecting

35th Conference on Neural Information Processing Systems (NeurIPS 2021).

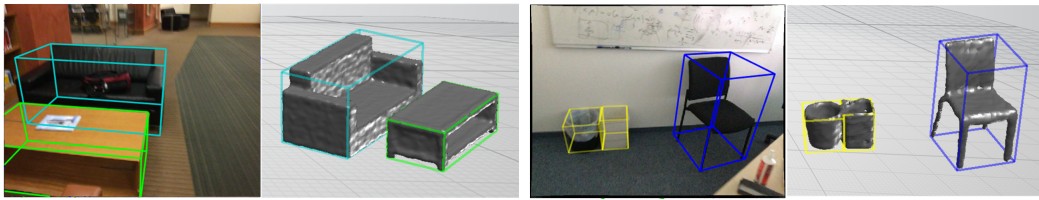

Figure 1: Given a single image as input, our proposed approach jointly predicts 3D object bounding boxes and surfaces.

2D features to 3D space for respective rendering or single object reconstruction purposes. Inspired by this line of works, we propose a novel voxel-based 3D detection and reconstruction framework for predicting 3D bounding boxes and surfaces of multiple objects from a single image (see Fig. 1).

Specifically, we first divide a 3D scene space into a regular grid of voxels. For each voxel, we assign 3D features by sampling from the image plane via a 2D-to-3D feature lifting operator and the known camera projection matrix. As multiple voxels can be projected to the same position, this leads to similar features along the camera ray and increased difficulty for downstream tasks. To remedy this, we use a positional encoding strategy to make our voxel features position-aware and more discriminative. Based on the intermediate voxel features, we carefully devise our detection and reconstruction modules. For detection, we introduce a novel CenterNet-3D detector head. Instead of formulating the 3D detection as 2D keypoint detection problem as conventional CenterNet-based methods [28, 29], each object is directly represented by its 3D keypoint. Predicting a class-specific 3D heatmap can show probabilities of 3D object centers in the pre-defined voxel space, leading to improved 3D center accuracy. For reconstruction, we propose a multi-level shape representation with two components: coarse-level occupancy representation and fine-level local PCA-SDF representation. The coarse-level voxel grid represents the whole 3D scene with continuous occupancy values. At a fine level, we represent the occupied voxels with a PCA-based signed distance function (SDF) by assuming that the local shapes of different voxels are similar either within an object instance, or across different objects.

In summary, the contributions of this work include:

◇ We propose a novel voxel-based 3D detection and reconstruction framework, which infers the 3D locations and 3D surfaces for multiple object instances with only a 2D image as input.

◇ We present a novel CenterNet-3D detector, where each object is represented by its center point in a partitioned 3D grid space. CenterNet-3D avoids estimating depth directly from image features, leading to increased detection performance.

◇ We propose a novel local PCA-SDF shape representation, which provides finer reconstruction and order of magnitude faster inference than SOTA local implicit function methods like DeepLS [30].

◇ We demonstrate the superiority of our method in multiple object 3D reconstruction and detection, as well as 3D shape representation. We assemble a 3D detection and reconstruction benchmark with 18, 000 real images, annotated with 3D models and bounding boxes of 19 object categories.

## 2 Related Work

**3D Scene Understanding and Single Object Reconstruction.** Tremendous efforts have been devoted to instance-level 3D scene understanding [7, 9, 18, 28, 31–37] over the last decade. However, most of these approaches estimate object orientation [34, 35, 38] or 3D bounding boxes [7–9, 39–45]. Since describing objects with boxes only offers a coarse information of 3D objects in images, the usage of 3D models as shape priors can complement and enrich 3D scene understanding. Yet, scene understanding at the instance level remains challenging due to the large number of objects with various categories. With the substantial growth in the number of publicly available 3D models, datasets such as ShapeNet [46] have allowed neural networks to train on the 3D shape reconstruction task from single images [47–52]. To further leverage real-world images in 3D modeling, as Liu *et al.* [53] propose a semi-supervised learning framework for generic objects. However, most of these methods estimate 3D shapes in the object-centric coordinate system, which differs from the shape prediction of multiple instances at scene-level 3D reconstruction – the focus of our work.

**Multiple Object 3D Reconstruction** A common characteristic amongst aforementioned single object 3D reconstruction approaches is that they usually treat objects as isolated geometries without considering the scene context, such as object locations, and instance-to-instance interactions. Recently, there is progress in multiple object 3D reconstruction. 3D-RCNN [18] exploits the idea of using inverse graphics to map image regions to the 3D shape and pose of object instances. The shape is represented by a simple linear subspace which limits its application for objects with large intra-class variability. Mesh R-CNN [21] augments Mask R-CNN [54] with a mesh predictions branch that estimates a 3D mesh for each detected in an image. Total3DUnderstanding [19] presents a framework that predicts room layout, 3D object bounding boxes, and meshes for all objects in an image based on the known 2D bounding boxes. However, these three methods first detect objects in the 2D image, and then *independently* produce their 3D shapes with single object reconstruction modules. This could be problematic when 3D boxes of objects intersect, such as a chair is pushed under a table.

Recently, CoReNet [22] performs multiple object reconstructions in a fixed $128^3$ voxel grid without recovering 3D position information in the world space. Points2Objects [23] combines a 3D object detector and shape retrieval to detect and reconstruct 3D objects from an image. However, it suffers from two limitations: 1) Its CenterNet-based 3D detector reasons 3D boxes directly in the 2D image domain, which is inherently challenging due to the lack of reliable depth cue. 2) Retrieval-based methods depend on the size and diversity of the pre-defined CAD model pool. Moreover, [22, 23] train on synthetic renderings, which limits their applicability to real-world scenarios. Instead of relying on 2D feature for 3D detection or reconstruction, we propose to learn a geometry and context preserving voxel feature representation, which is well suited for 3D detection and reconstruction. Moreover, we validate our method on real-world images from 19 object categories.

**Local Shape Priors** Many neural architectures are proposed to model 3D objects via geometric representations, *e.g.*, point clouds [55], meshes [16, 56], voxels [13, 57], or implicit functions [15, 58–60]. Recently, neural implicit functions have demonstrated their effectiveness by encoding geometry in latent vectors and network weights, which parameterize surfaces through level-sets. Instead of an object-level representation, some follow-up works learn patch-level or primitive-level representations of surfaces, *e.g.*, PatchNet [61], CvxNet [62], BSP-Net [63]. To further leverage local geometric priors, another line of works learn implicit geometry on sparse regular [64, 65] or 3D voxel grids [30, 66]. A latent code of each voxel is responsible for representing implicit geometry in a small neighborhood, enabling fine-grained reconstruction. However, these methods often suffer from inefficient inference as each point needs a forward pass through of the implicit function network. Instead, we build a local PCA-SDF shape representation, which represents each local shape as a linear combination of implicit volumetric prototypes, leading to finer details and order of magnitude faster inference than prior works. Similar eigenanalysis of SDF has been applied for either global shape representation [67] or geometry compression [68, 69]. However, none of them develop their algorithms from our perspective of local shape priors, which is motivated from the assumption that local shapes at the voxel level share similarities.

# 3 Methodology

We illustrate the overall architecture of our method in Fig. 2, which consists of three key modules: i) 3D voxel feature learning; ii) CenterNet-3D detector; and iii) Coarse-to-fine 3D reconstruction.

## 3.1 3D Voxel Feature Learning

Our network learns to produce a compact 3D voxel feature representation of the image with *complementary* supervision from both 3D detection and reconstruction, which enables rich 3D context and geometric information and allows the two tasks to benefit each other. We first define a 3D grid $\mathbf{V} \in \mathbb{R}^{X \times Y \times Z}$ by partitioning of the scene space into voxels $\mathbf{V}_i$ with a voxel size of $r$. The 3D grid size, $Xr \times Yr \times Zr$, is set based on the minimum volume of the scenes in the width, height and length dimensions, that can encompass all annotated instances in the dataset.

**Feature Extraction.** We utilize a convolutional feature extractor to generate a hierarchy of multi-scale 2D feature maps. Specifically, the input to the feature extraction network is a RGB image $\mathbf{I} \in \mathbb{R}^{W_I \times H_I \times 3}$, where $W_I$ and $H_I$ are the image size. The convolutions are followed by down scaling the input, creating growing receptive fields and resulting in $D$-channel multi-scale 2D feature maps $\mathbf{F} \in \mathbb{R}^{W_F \times H_F \times D}$. $W_F$ and $H_F$ are the width and height of feature $\mathbf{F}$.

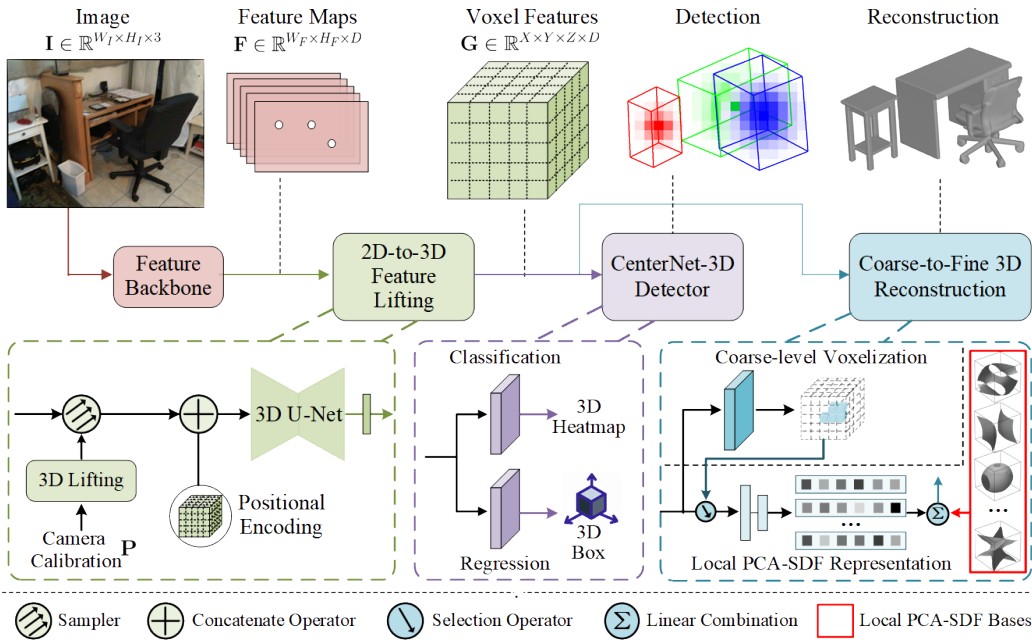

Figure 2: **Overview of our approach.** The proposed joint framework is composed of three key modules: 3D voxel feature learning (consists of feature backbone and 2D-to-3D feature lifting), CenterNet-3D detector, and coarse-to-fine 3D reconstruction. 2D feature maps are first generated from input image $\mathbf{I}$, which are back-projected into voxel features $\mathbf{G}$ using a known camera projection matrix $\mathbf{P}$. The voxel features serve for our novel 3D object detection and reconstruction.

**Lifting 2D Features to 3D.** The lifting layer back-projects 2D feature maps $\mathbf{F}$ into 3D voxel space, resulting in *initial* 3D voxel features $\mathbf{G}$. Formally, the objective of the lifting operator is to populate the 3D voxel features $\mathbf{G}(x, y, z)$ with the projected 2D features $\mathbf{F}\{(u, v)\}$, where $\{\cdot\}$ denotes bilinear interpolation on the 2D feature maps. We assume a full perspective camera model. Any voxel center $(x, y, z)$ can be projected to image plane via a camera projection matrix $\mathbf{P} \in \mathbb{R}^{3 \times 4}$: $[u \cdot d, v \cdot d, d]^T = \mathbf{P}[x, y, z, 1]^T$. Here $u$ and $v$ are the 2D position of the projection and $d$ is its depth from the camera. The resulting voxel features $\mathbf{G} \in \mathbb{R}^{X \times Y \times Z \times D}$ provide a scene representation that is free from the effects of perspective projection.

**3D Voxel Features Aggregation.** The lifting mechanism we use is similar to the one in [24, 25, 27], which has a major weakness that all voxels along a camera ray will receive the same 2D feature. This feature smearing issue increases the difficulties of 3D detection and reconstruction. To mitigate this issue, we propose to employ the *positional encoding* (PE) [70] strategy that adds 3D voxel center position to the voxel features: $\mathbb{R}^{X \times Y \times Z \times D} \xrightarrow{PE} \mathbb{R}^{X \times Y \times Z \times (D+3)}$, which helps the voxel features to be more discriminative and position-embedded. We further utilize a 3D convolutional hourglass (U-Net) network [71], comprised of a series of down- and upsampling convolutions with skip connections, to integrate both local and global information. The final voxel features are thus $\mathbf{G} : \mathbb{R}^{X \times Y \times Z \times (D+3)} \xrightarrow{U-Net} \mathbb{R}^{X \times Y \times Z \times (D+3)}$. This voxel features serve as the cornerstone for the downstream tasks of 3D detection and reconstruction.

## 3.2 Monocular CenterNet-3D Detector

Conventional CenterNet-based [28] monocular 3D detection methods such as Points2Objects [23] formulate 3D detection as a projected 2*D keypoint* detection problem. In contrast, we propose a novel CenterNet-3D detection head, where each object is directly represented by its 3*D keypoint*. Then 3D properties such as object size and orientation can be intuitively inferred from the 3D voxel features at the center location by a regression branch. Compared to [23], CenterNet-3D avoids estimating depth values of 3D boxes directly from 2D features, leading to improved detection accuracy.

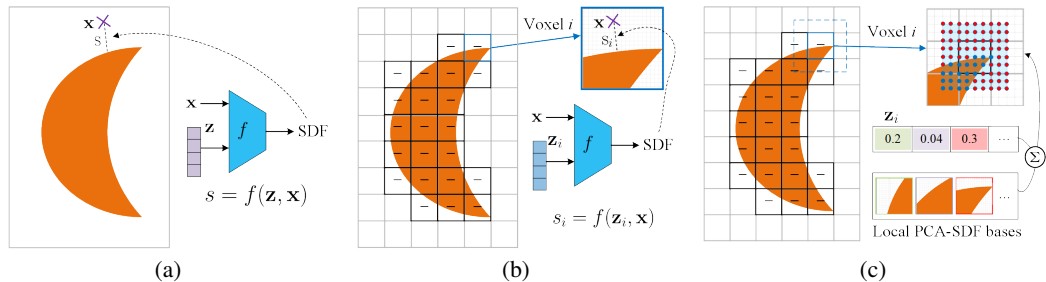

Figure 3: 2D examples of (a) DeepSDF [59], (b) DeepLS [30], and (c) our local PCA-SDF shape representation. DeepSDF describes the surfaces with global shape codes. The SDF function $f$ in DeepLS outputs a scalar value conditional on the local latent code $z_i$ and local coordinate $x$. However, its inference is computationally expensive since it requires forward pass through $f$ for every $x$. Our shape representation consists of coarse-level voxelization and fine-level local PCA-SDF. The coarse-level voxelization holistically represents the whole 3D surface with binary values. To further represent fine-level surfaces, we propose a novel local PCA-SDF model, representing any occupied voxel as a linear combination of regular SDF function bases, which enables a more efficient and accurate representation than DeepLS.

**3D Keypoint Branch.** The 3D keypoint branch takes the voxel features $\mathbf{G}$ as input and predicts a *3D heatmap* $\mathcal{Y} \in \mathbb{R}^{X \times Y \times Z \times C}$ (see Fig. 2), where $C$ is the number of object categories. The values of each voxel in $\mathcal{Y}$ indicates how likely the 3D centroid of a certain object category exists at the voxel center. By computing the local maxima and filtering via a threshold, we obtain a preliminary estimation of the 3D centroids, denoted as $\tilde{\mathbf{c}}_{3d} = [x_c, y_c, z_c]^T$. To remedy the discretization error of voxels, the regression branch additionally predicts a local offset to $\tilde{\mathbf{c}}_{3d}$, which is discussed next.

**Regression Branch.** The regression branch predicts the essential properties to construct a 3D bounding box for each voxel of the 3D heatmap. We parameterize a 3D box as the prior works [8, 19] and set up the world system located at the camera center with its vertical ($y$-) axis perpendicular to the floor and its forward ($z$-) axis toward the camera, such as the pitch and roll angles could be included in the camera pose. Specifically the 3D box is encoded as a 8-tuple $\tau = [\delta_{x_c}, \delta_{y_c}, \delta_{z_c}, \delta_h, \delta_w, \delta_l, \sin\theta, \cos\theta]$. Here $\Delta\mathbf{c}_{3d} = [\delta_{x_c}, \delta_{y_c}, \delta_{z_c}]^T$ denotes the 3D center offset compensating voxel discretization. $[l, h, w]^T = [\bar{l} \cdot e^{\delta_l}, \bar{h} \cdot e^{\delta_h}, \bar{w} \cdot e^{\delta_w}]^T$ represents the object size, where $[\bar{l}, \bar{h}, \bar{w}]^T$ is a pre-calculated category-wise average box size, $[\delta_h, \delta_w, \delta_l]$ represents the corresponding transformations. $\theta$ denotes the rotation angle around $y$-axis. Here the network estimates the vectorial representation of rotation angle $\theta$ [72]. The output size of the regression branch is thus $X \times Y \times Z \times 8$. Given the outputs of keypoint and regression branches, the 3D bounding box $\mathcal{B} \in \mathbb{R}^{3 \times 8}$ can be restored as 8 corners:

$$\mathcal{B} = R_\theta \begin{bmatrix} \pm l/2 \\ \pm h/2 \\ \pm w/2 \end{bmatrix} + \mathbf{c}_{3d}, \quad \mathbf{c}_{3d} = \tilde{\mathbf{c}}_{3d} + \Delta\mathbf{c}_{3d}, \tag{1}$$

where $R_\theta \in \mathbb{R}^{3 \times 3}$ is the rotation matrix.

### 3.3 Coarse-to-Fine 3D Reconstruction

Our reconstruction module is based on a coarse-to-fine shape representation, which consists of two components: coarse-level voxelization and fine-level local PCA-SDF.

**Coarse-Level Voxelization.** Based on the extracted 3D voxel features $\mathbf{G}$, we first estimate a coarse-level voxelization $\tilde{\mathcal{V}}$ by a specific branch. The coarse-level voxelization holistically represents the whole 3D surface with binary occupancy values, where the unoccupied voxels cover "air" in the scene, and occupied voxels can be either fully occupied ones inside the object, or voxels intersecting with the object's surface. For the occupied voxels of both types, we further reconstruct a fine-level local shape via the local PCA-SDF.

**Local PCA-SDF Shape Representation.** Recent works such as DeepSDF [59] aims to learn global implicit functions to represent shapes (see Fig. 3(a)). However, representing the entire objects with a single latent code often results in loss of details, which limits its application for scene-level object reconstruction. DeepLS [30] represents 3D surfaces by a set of independent latent codes on a regular

grid (see Fig. 3(b)). Each latent code $\mathbf{z}_i$, concatenated with any point location $\mathbf{x}$, can be decoded into a SDF value $s_i$ by the learned implicit network $f$: $s_i = f(\mathbf{z}_i, \mathbf{x})$. However, this shape representation has two limitations. i) Inference is inefficient (in the order of seconds) since every point of a test voxel (*e.g.,* $256^3$ points) is required to be sent to $f$ for SDF calculation, making it unsuitable for real-time applications. ii) Our key observation is that local voxels, either within an object instance or across different categories, share similar local shapes, *e.g.*, voxels across a table's surface all have planar shapes. However, DeepLS treats voxels as independent training samples, without fully leveraging such local shape priors in training. To address these issues, we propose a novel local PCA-SDF shape representation, which represents each voxel shape as a linear combination of a set of implicit volumetric prototypes, leading to significantly finer reconstruction and $10\times$ faster inference speed than DeepLS (see Tab. 4).

Formally, as shown in Fig. 3(c), for each occupied voxel $\mathbf{V}$, we define a regular lattice $\mathbf{q} \in \mathbb{R}^{k \times k \times k \times 3}$ and compute their SDFs $\mathbf{s} \in \mathbb{R}^{k \times k \times k \times 1}$ toward the surface. By collecting SDFs of $N_S$ occupied voxels from the training surfaces, we apply Principal Component Analysis (PCA) to find $l_B$ ($l_B <<$ $k^3$) local shape bases, $\mathcal{S}_B \in \mathbb{R}^{k \times k \times k \times l_b}$. As such, given the learned $\mathcal{S}_B$ and the latent code $\mathbf{z}_i$, any local shape $\mathcal{S}_i$ of the underlying surface can be implicitly represented by $\mathcal{S}_i = \mathcal{S}_B \mathbf{z}_i$. The latent code $\mathbf{z}_i$ for $\mathcal{S}_i$ can be generated by the corresponding voxel feature $\mathbf{G}_i$ via $\mathbf{z}_i = \mathrm{MLP}(\mathbf{G}_i)$. MLP is a mapping network, implemented with two fully-connected layers. By combining the contributions of all the occupied voxels, we can infer a global iso-surface from the SDF field. Similar to DeepLS [30], we apply a 1.5 times receptive field strategy to mitigate the inconsistent surface predictions at the voxel boundaries (Fig. 3(c)). Accordingly, during inference, we could apply average pooling to combine the SDF values for the boundary area. It is worth mentioning that our local PCA-SDF also allows reconstruction at resolutions higher than the one used during training by simply applying trilinear interpolation on the learned local shape bases $\mathcal{S}_B$.

### 3.4 Loss Functions and Implementation Details

The training data of one example consists of RGB image $\mathbf{I}$, ground-truth 3D bounding boxes $\mathcal{B}^*$ of objects, coarse-level voxelization $\mathcal{V}^*$, and a set of regular SDF pairs $\{(\mathrm{idx}_j, \mathbf{s}_j)\}_{j=1}^{K}$ sampled from the surface. Here, each set of SDFs $\mathbf{s}_j \in \mathbb{R}^{k \times k \times k}$, $\mathrm{idx}_j$ is the $\mathrm{idx}_j$-th voxel of the holistic grid. During training, we jointly optimize the parameters of 2D feature extraction network, 3D U-Net, detection and reconstruction modules by minimizing three losses: 3D keypoint classification loss $\mathcal{L}_{cls}$, regression loss $\mathcal{L}_{reg}$, and 3D reconstruction loss $\mathcal{L}_{recon}$, *i.e.,*

$$\mathcal{L} = \mathcal{L}_{cls} + \mathcal{L}_{reg} + \mathcal{L}_{recon}. \tag{2}$$

**Loss Functions.** We generate the target heatmaps $\mathcal{Y}^*$ by splatting the ground truth 3D center points using a Gaussian kernel (please refer to **Supp** for details). If two Gaussians of the same class overlap, we take the element-wise maximum. The 3D keypoint branch is trained with a penalty-reduced focal loss [28,73] in a point-wise manner on the 3D heatmap.

$$\mathcal{L}_{cls} = \frac{-1}{N} \sum_{xyzc} \begin{cases} (1 - \mathcal{Y}_{xyzc})^\mu \log(\mathcal{Y}_{xyzc}) & \text{if} \quad \mathcal{Y}^*_{xyzc} = 1 \\ (1 - \mathcal{Y}^*_{xyzc})^\sigma (\mathcal{Y}_{xyzc})^\mu \log(1 - \mathcal{Y}_{xyzc}) & \text{otherwise} \end{cases} \tag{3}$$

where $N$ is the number of objects per image, $\mu = 2$ and $\sigma = 4$ are hyper-parameters of the focal loss.

We define the 3D bounding box regression loss as the $L_1$ distance between the predicted transform $\mathcal{B}$ and the ground truth $\mathcal{B}^*$: $\mathcal{L}_{reg} = \frac{1}{N} ||\mathcal{B} - \mathcal{B}^*||_1$. The reconstruction loss consists of cross-entropy classification loss $\mathcal{L}_v$ for coarse-level voxelization and fine-level SDF regression loss.

$$\mathcal{L}_{recon} = \mathcal{L}_v(\mathcal{V}, \mathcal{V}^*) + \sum_{j}^{K} ||\mathcal{S}_B \mathbf{z}_j - \mathbf{s}_j||_2^2, \quad \mathbf{z}_j = \mathrm{MLP}(\mathbf{G}_{\mathrm{idx}_j}). \tag{4}$$

**Implementation Detail.** We use a ResNet-34 network as our 2D feature extractor. We extract features immediately before the final three downsampling layers, resulting in a set of feature maps at $1/8$, $1/16$, $1/32$ scales of the input resolution. We then resize them to the original size of the input images via bilinear interpolation. Convolutional layers with $1 \times 1$ kernels are used to map these feature maps to a common channel size of $64$. The set of feature maps is summarized as $\mathbf{F}$ before

Table 1: **Multiple object reconstruction comparison.** We report per-class and mean IoU over all classes, and class-agnostic global IoU on $128^3$ voxel grid.

| Method | ShapeNet-triplets | | | | | | | | ShapeNet-pairs | |
|---|---|---|---|---|---|---|---|---|---|---|
| | bottle | bowl | chair | mug | sofa | table | mean | global | mean | global |
| CoReNet [22] | 61.8 | 36.2 | 30.1 | 48.0 | 52.9 | 34.8 | 43.9 | 49.8 | 43.1 | 52.7 |
| Points2Objects [23] | **63.5** | 30.2 | 18.9 | 41.5 | 44.5 | 19.8 | 36.4 | 44.7 | – | – |
| Proposed | 63.3 | **38.5** | **31.8** | **51.7** | **54.3** | **36.1** | **46.0** | **52.3** | **46.7** | **55.1** |

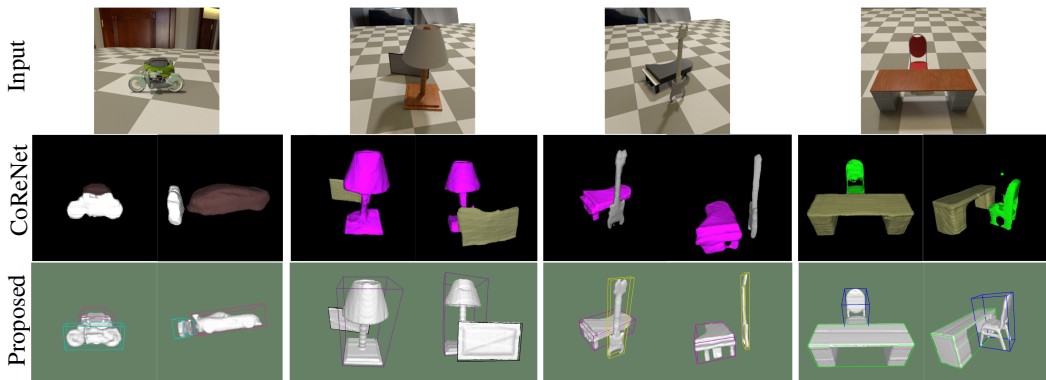

Figure 4: **Qualitative results on ShapeNet-triplets dataset.** We compare to CoReNet [22] in two different viewpoints. Our model more accurately reconstructs details and hallucinates occluded parts.

passing to 3D lifting layers. For the main experiments, we train our models with a batch size of 8 on a GTX 1080Ti GPU for 200 epochs. The learning rate is set at $5 \times 10^{-4}$ and drops at 50 and 100 epochs by a factor of 10. For more details, please refer to Sec. 4 or **Supp**.

# 4  Experiments

## 4.1  Multiple Object Detection and Reconstruction on ShapeNet-pairs and -triplets

**Datasets.** Following the experimental setting of CoReNet [22] and Points2Objects [23], we evaluate multiple object detection and reconstruction on *ShapeNet-pairs* and *ShapeNet-triplets* datasets [22]. These datasets contain $256 \times 256$ photorealistic renderings of either pairs or triplets of ShapeNet [46] objects placed on a ground plane with random scale, rotation, and camera viewpoint. The *ShapeNet-pairs* has several pairs of object classes: bed-pillow, bottle-bowl, bottle-mug, chair-table, display-lamp, guitar-piano, motorcycle-car and sofa-table, which contains $365,600$ images on trainval and $91,200$ on test. The *ShapeNet-triplets* is with bottle-bowl-mug and chair-sofa-table, which includes $91,400$ on trainval and $22,000$ on test.

**Experimental Settings.** In this experiment, we set a voxel grid of size $X \times Y \times Z = 40 \times 25 \times 25$ ($r = 0.1$), which is sufficient to enclose all objects in the datasets. We randomly select 200 surfaces from the training set to generate $N_S \approx 200,000$ occupied voxels. In this experiment, we set $k = 11$, $l_B = 64$, $C = 6$ (ShapeNet-triplets) or $C = 14$ (ShapeNet-pairs). We compare with SOTA methods for multiple object reconstruction: CoReNet [22] and Points2Objects [23]. As CoReNet doesn't perform 3D detection, we only compare with Points2Objects on detection. Following [23], we use mean average precision (mAP) as the detection metric with 3D box intersection-over-union (IoU) thresholds 0.25 and 0.5. Following [22,23], the metric for reconstruction is IoU on a $128^3$ voxel grid.

**Results.** We first report the 3D detection results. Our method achieves a higher detection accuracy than Points2Objects [23]: **51.5**% *vs.* $48.6$% (threshold@0.5) and **80.3**% *vs.* $77.2$% (threshold@0.25), which demonstrates that voxel features perform better than image-based features for monocular 3D detection. For reconstruction, we report the mean over the per-class IoU, as well as the global IoU of all object instances within a scene, which does not concern predicted class labels. As compared in Tab. 1, our method significantly outperforms two SOTA baselines on both datasets. On ShapeNet-triplets, our method achieves relative $5.0$% global IoUs gains while $4.8$% mean IoUs gains, which

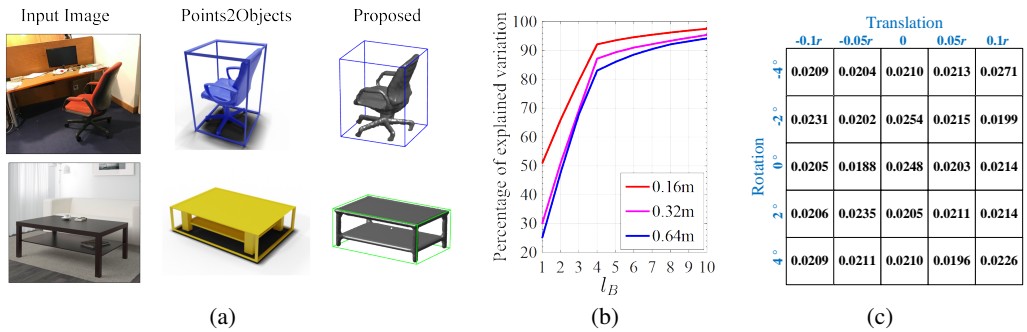

(a)            (b)            (c)

Figure 5: (a) Qualitative comparison on Pix3D. Our reconstructions closely match objects' genuine shape, *e.g.,* the table legs and chair arms. (b) Explained variation of our PCA-SDF representation with three voxel sizes $r$. (c) Reconstruction errors (Chamfer Distance-$L2$) of PCA-SDF w.r.t. translated and rotated 3D shapes.

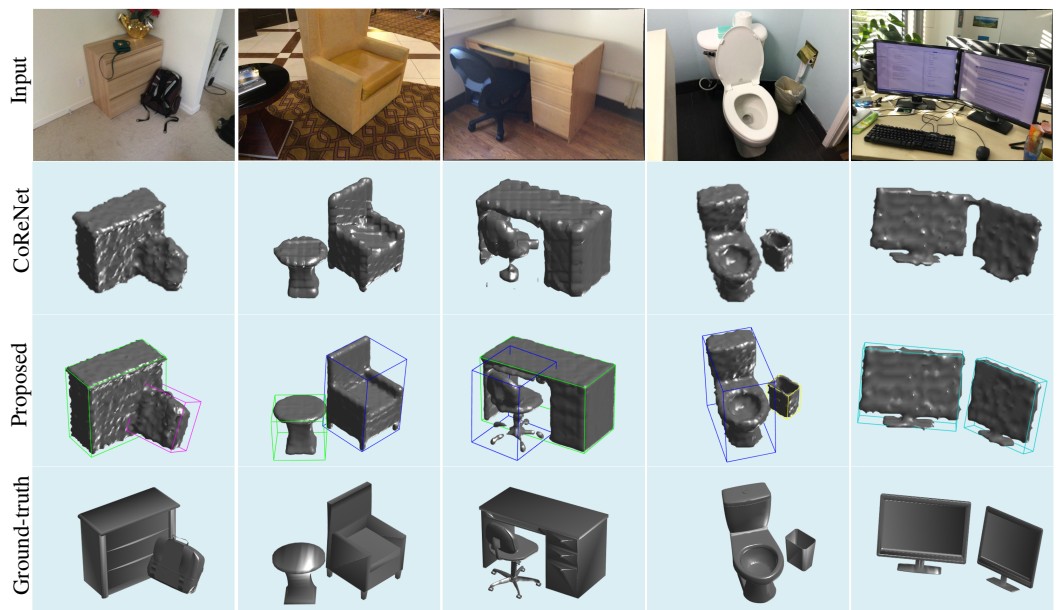

Figure 6: **Qualitative results on real images from ScanNet-MDR.** Our reconstructions closely match the objects than CoReNet [22]. Moreover, our method performs better for reconstruction of the truncated objects.

indicates that our model performs well on reconstructing the overall shapes of objects. Qualitative results of detection and reconstruction are shown in Fig. 4.

## 4.2 Single Object Reconstruction on Pix3D

We further compare to CoReNet [22] and Points2Objects [23] on the real image database, Pix3D [74] in the same protocol (splits $S_1$ and $S_2$) as in [21]. In this experiment, we train our model with the same experimental setting and the same pre-computed PCA-SDF bases as in Sec. 4.1. On average IoU over all 9 object classes, we achieve **38.6**% $vs.$ 34.1% (CoReNet) $vs.$ 33.3% (Points2Objects) on $S_1$ and **28.6**% $vs.$ 26.3% (CoReNet) $vs.$ 23.6% (Points2Objects) on $S_2$. The results demonstrate that our approach improves over baselines on real images. Qualitative results are shown in Fig. 5(a).

## 4.3 Multiple Object Detection and Reconstruction on ScanNet-MDR

**Dataset.** Since there is no benchmark providing both 3D CAD models and 3D bounding boxes for multiple objects within a single real image, we assemble a dataset with $18,000$ real images from the

Table 2: Comparisons of 3D object detection and reconstruction on ScanNet-MDR dataset. [Key: ①=CenterNet, ②=CenterNet-3D], [①=DeepLS, ②=Local PCA-SDF]

| Method | Detection | | Recon. | | Evaluation | |
|---|---|---|---|---|---|---|
| | ① | ② | ① | ② | mAP (@0.15) | IoU |
| CoReNet [22] | | | – | – | – | 35.2 |
| Proposed-1 | | | | ✓ | – | 36.5 |
| Proposed-2 | | ✓ | | | 20.7 | – |
| Proposed-3 | ✓ | | | ✓ | 19.5 | 36.9 |
| Proposed-4 | | ✓ | ✓ | | 20.0 | 35.4 |
| Proposed-w/o PE | | ✓ | | ✓ | **21.2** | **37.2** |
| Proposed | | ✓ | | ✓ | 22.8 | 38.2 |

Table 3: Effect of the voxel size $r$ and latent code size $l_B$ in detection and reconstruction on ScanNet-MDR dataset (mAP/IoU), and inference time per image.

| $r$ \ $l_B$ | 16 | 32 | 64 | Time (ms) |
|---|---|---|---|---|
| 0.64m | 17.5/ 30.7 | 17.7/ 31.1 | 18.1/ 31.4 | 44.2 |
| 0.32m | 19.1/ 34.1 | 18.8/ 34.8 | 18.6/ 35.3 | 55.6 |
| 0.16m | 21.9/ 37.1 | 22.2/ 37.6 | **22.8**/ **38.2** | 78.1 |

ScanNet [75], termed ScanNet Monocular Detection and Reconstruction (ScanNet-MDR) dataset. For each object in an image, its CAD model is produced by [76]. We then generate the corresponding 3D bounding box label and camera calibration matrix. Unlike the ShapeNet-pairs or ShapeNet-triplets datasets, all the 3D objects in ScanNet-MDR are at absolute scale. Additionally, this dataset contains greater diversity including 19 object categories (bag, basket, bathtub, bed, bench, bookshelf, cabinet, chair, display, file, lamp, microwave, piano, printer, sofa, stove, table, trash and washer). We split the data into $80\%$ for training and $20\%$ for testing.

**Experimental Settings.** In this dataset, we use a voxel grid of size $10.28 \times 3.2 \times 6.4m$ ($X = 64, Y = 20, Z = 40, r = 0.16m$), a minimum to encompass all annotated 3D objects in the dataset. The PCA-SDF is pre-computed with $N_S \approx 560,000$ occupied voxels from 200 training surfaces. We set $k = 17, C = 19$ and $l_B = 64$. For comparison, we train CoReNet [22] using the released code on our training data. We use mAP with 3D box IoU threshold of $0.15$ as detection metric [19], and global 3D IoU on a $128^3$ voxel grid as reconstruction metric.

**Results and Ablation Studies.** We report detection and reconstruction results on the testing set. As shown in Tab. 2, our method significantly improves over CoReNet [22] on 3D reconstruction and advances the ablated versions on both detection and reconstruction. Qualitative results are shown in Fig. 6. CoReNet, as an image-to-voxel reconstruction network without special design for feature smearing issue, cannot handle truncated objects in the input image.

*Joint Framework vs. Separate Modules.* Moreover, Tab. 2 shows the ablation results of our models without detection (Proposed-1) or reconstruction (Proposed-2) modules, where one can conclude that joint framework in this work performs better than solving either task exclusively.

*CenterNet-3D and PCA-SDF.* To further validate the effectiveness of the proposed CenterNet-3D detector over the conventional CenterNet, we train a model (Proposed-3) by combining the reconstruction module with the conventional CenterNet, which formulates 3D detection as a problem of 2D keypoint detection directly from the pixel-based image features. As compared in Tab. 2, our model outperforms Proposed-3 in both detection and reconstruction. To compare PCA-SDF with DeepLS [30] in the joint detection and reconstruction framework, we train a model (Proposed-4) by using DeepLS representation as our fine-level reconstruction module. Tab. 2 shows that both detection and reconstruction performances are worse than ours.

*Effect on Voxel Size $r$ Latent Code Size $l_B$.* The validity of local shape pair, expressed by PCA-SDF in our work, depends on the voxel size. For instance, we show the percentage of explained variation for three voxel sizes in Fig. 5(b). As larger voxel sizes are used, the first few bases could explain less variation, due to the diminished local shape similarity among larger voxels, *i.e.*, weakened local shape prior. This is also validated by the ablation of voxel size $r$ and latent code size $l_B$ in Tab. 3, where larger voxel sizes lead to lower detection and reconstruction accuracies. On the other hand, a larger latent code size results in better representation power (Fig. 5(b)) but not necessarily reconstruction, since it imposes a more challenging task for the network to predict a higher-dim code.

*Effect on Positional Encoding.* To investigate the effect of positional encoding operator on 3D detection and reconstruction, we retrain a model without the positional encoding (Proposed-w/o PE). As compared in Tab. 2, both detection and reconstruction accuracies are worse than ours, which indicates that the positional encoding indeed enhances the voxel feature representation.

Table 4: Comparison of reconstructing 3D shapes from ShapeNet test set, evaluated by Chamfer Distance-$L2$ (multiplied by $10^3$). PCA-SDF achieves higher accuracy and efficiency than DeepLS even with fewer decoder and representation parameters. Decoder para. refer to the decoder network parameters for DeepSDF or DeepLS, and PCA bases for our PCA-SDF. [Key: **Best**, **Second Best**]

| Method | Chair | Plane | Table | Lamp | Sofa | Mean | Unseen | #Decoder Para. (M) | #Represent. Para. (K) | Inference Time (s) |
|---|---|---|---|---|---|---|---|---|---|---|
| DeepSDF [59] | 0.204 | 0.143 | 0.553 | 0.832 | 0.132 | 0.372 | - | 1.8 | **0.3** | 6.9626 |
| DeepLS [30] ($l_B$=125) | 0.030 | 0.018 | 0.032 | 0.078 | 0.044 | 0.040 | - | 0.05 | 4096 | 0.8081 |
| PCA-SDF ($l_B$=32) | 0.031 | 0.016 | 0.033 | 0.035 | 0.032 | 0.029 | 0.111 | **0.02** | 1049 | **0.0126** |
| PCA-SDF ($l_B$=64) | **0.027** | **0.012** | **0.030** | **0.027** | **0.030** | **0.025** | 0.059 | 0.05 | 2097 | 0.0129 |
| PCA-SDF ($l_B$=125) | **0.026** | **0.010** | **0.029** | **0.016** | **0.029** | **0.022** | **0.028** | 0.09 | 4096 | 0.0132 |

*Computation Time.* Tab. 3 validates our inference time per image with different voxel sizes on a GTX 1080Ti GPU. Since $l_B$ does not affect the runtime much, we show the average time across three $l_B$.

### 4.4 3D Shape Representation Power of PCA-SDF

While PCA-SDF has demonstrated its advantage in 2D to 3D reconstruction, this is rooted from its ability in representing 3D shapes. To quantify its 3D shape representation power, we design the following experiment and compare with DeepSDF [59] and DeepLS [30], without involving 2D image inputs. Following the setting of [30], we utilize $1,000$ ShapeNet shapes ($200$ each from $5$ categories) to compute our PCA-SDF bases. Each 3D shape is split by a $32 \times 32 \times 32$ grids ($r = \frac{1}{32}$). During training, both DeepSDF and DeepLS optimize the latent codes and decoder weights through backpropagation, to best represent the training shapes. In inference, decoder weights are fixed, and the optimal latent code is estimated given a testing shape. In contrast, we compute the latent codes whose multiplication with PCA-SDF bases can best approximate the ground-truth SDF.

We evaluate 3D shape reconstruction accuracy on various categories. As shown in Tab. 4, PCA-SDF has lower reconstruction error than DeepLS, even with a smaller number of representation parameters. Moreover, our inference is $10\times$ more efficient (infer at $256^3$ resolution) which meets the real-time requirement for downstream tasks.

To study whether the PCA-based representation is sensitive to tiny geometry perturbation, we apply minimal translation ($\pm 0.1r$, $\pm 0.05r$) and rotation ($\pm 4°$, $\pm 2°$) to testing surfaces of the $5$ categories and evaluate surface reconstruction error on these data, while no data augmentation was applied to the training data of shape bases computing. As shown in Fig. 5(c), the error is stable in a very small range, which illustrates that PCA-SDF is robust to translation and rotation variations.

*Generalization to Unseen Category.* In order to investigate the generalization of PCA-SDF, we design an experiment to compute PCA-SDF from a single category ($200$ shapes), and reconstruct 3D shapes from the other four unseen categories. We repeat the training/testing $5$ times across $5$ categories. As reported in Tab. 4, PCA-SDF trained on unseen categories achieves a comparable performance ($0.028$ *vs.* $0.022$) with the one trained on seen categories when $l_B = 125$, which indicates that local 3D shapes at the voxel level are indeed similar to each other, even across different categories.

## 5 Conclusion

We present a voxel-based 3D detection and reconstruction framework for predicting 3D bounding boxes and shapes of multiple objects from a single image. Specifically, we first learn a regular grid of 3D voxel features for the input images. Based on the voxel features, we devise a novel CenterNet-3D detector to detect and regress 3D bounding boxes in the 3D space. With a coarse-level voxelization and a fine-level local PCA-SDF representation, our reconstruction module provides highly efficient and accurate reconstructions. The comprehensive experiments show the superiority of the proposed method in 3D detection and reconstruction, as well as shape representation power. The same as CoReNet and Points2Objects, one limitation of our approach is that it requires the camera calibration matrix as input which might limit its application to real images. Therefore, one future direction is to invest the necessity of this requirement and/or integrate with auto-calibration methods.

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
