# OpenReview forum: "Voxel-based 3D Detection and Reconstruction of Multiple Objects from a Single Image"
_NeurIPS.cc/2021/Conference — NeurIPS 2021 Poster_

### Official Review · Reviewer_gf8G · 2021-07-07

**Rating:** 8
**Confidence:** 4

**Summary:**

The paper proposes a technique for joint detection and reconstruction of objects from images. The core idea is to express the signed distance function (used for reconstruction) within a set of pre-trained linear bases, and regress the bases coefficients by a deep network. The effectiveness of the solution is evaluated on a number of datasets against recently published work, and shows good performance.

**Ethical Concerns:**

None.

**Limitations And Societal Impact:**

Not sure, and not sure I should care for a paper of this type.

**Main Review:**

Overall, I think the technique is solid and easy to understand, the results very competitive against SOTA (with a simpler technique), and the writing simply **excellent** (congrats! the introduction was the *only* section I did not enjoy reading). I am willing to raise the score by one point if my questions are answered properly, and I do not score the paper even higher, simply because I have not personally worked on all of the datasets this work is evaluated on (I would like to see the other reviewer's comments before taking a championing stance).

## Main questions (Q):
1. my main concern is the overall case of DeepLS vs. PCA-SDF. While without doubt the latter should be more computationally performant, I am skeptical that the accuracy (or latent space) of the former to not be more expressive. Aside from experiments, the paper lacks a theoretical justification of why this should be the case, and I would like to hear the authors' opinion in regards. For example, the recent NGLOD and ACORN papers use *tiny* decoders, and they represent geometry to very high accuracy?
1. the use of positional encoding in the UNet structure is one of the main differences w.r.t. past works (I am aware of), and I am simply amazed it was not quantitatively ablated? I only saw qualitative comparisons in Suppl/Fig.2. I would expect the method to collapse without this enabled in the multi-object setting?
1. Why euler angles and not quaternions, expmaps, or other vector representations? euler is especially a challenge as the representation is non-periodic (0==360). How can this not be an issue?
1. is reconstruction performed within each object's bounding box domain, or at the scene level (I assume the latter)? regardless of the answer, picking one begs you should compare against the other in terms of reconstruction accuracy? Why was this not done?
1. if an image is captured in the wild, knowing where to place the grid becomes a chicken and egg problem (you could estimate the grid position from detections, but detections are derived from the grid). Shouldn't this limitation be clearly acknowledged?
1. how finnicky is thresholding the detection maps? And if this becomes noisy, wouldn't this require some form of maximal suppression?
1. Sec 4.2: You are using the Pix3D splits from MeshRCNN, yet provide no quantitative comparisons to this paper... can you justify?
1. you mention "truncated objects" and that other methods have "no special design for feature smearing"... does yours have it? and can you spell out the issue?

## Technical details (T):
1. I am pretty sure in (1) you want to write everything in homogeneous coordinates.. otherwise the math is actually wrong (e.g. $c_3d$ becomes a scale rather than a translation)
1. You mention interpolation in L199, but I do not recall seeing any results/ablations in regards? Also note that NSVF and NGLOD both allow for this?
1. in (3) you employ the L1 norm for projecting the signal to the PCA basis in the loss, but PCA uses the L2 norm in its optimization, have you tried L2? Doesn't L1 result in suboptimal performance due to the mismatch?
1. L213: the L1 norm of a matrix is the maximum absolute column sum... pretty sure that this is not what you meant to say, are you referring to the Frobenius norm?

## Further comments (C)

1. Title: not sure that this paper is voxel based, as when I think of voxels I do not think of implicit functions... consider changing your title to reflect this?
1. L32-37 should be moved to the related works. It communicated very little to me as a reader
1. L112-113: not sure this sentence is true, I think it is more proper to say that this is not used within a deep network for inference task (it was used as a prior for compression)?
1. L123-126: it would be good to acknowledge past work from which you inherited this design, I recall learning it from  "learnable triangulation of human pose" (LTUP), but I am sure there was an earlier paper...
1. L127-144: same as above, this is almost identical to LTUP, with the exception of positional encoding?
1. L161: residual with respect to what? it was only mentioned later on in the paper, and you should mention it here
1. L155: it would be good to note that, compared to past works such as KeypointNet or LTUP this is performed because you need multi-modal instance detection.
1. L170: voXELization
1. L188: q is defined and never used (this is also typically called a `meshgrid`)
1. L189: what does "toward the surface" mean?
1. L209: please add a line to mention why this focal loss is needed?
1. L215: typo "without as"
1. L303: how? by the classical inner product of the lexicographically-unrolled SDF with the PCA projection matrix? (hint: cite or math, your choice)
1. L34: please avoid undeclared acronyms at all costs.. BEV? Bird Eye View? maybe...

**Time Spent Reviewing:**

3

---

> ### Author Response · Authors · 2021-08-10
> **Response to Reviewer gf8G**
>
>
> We thank the reviewer for the detailed and thorough feedback. The reviewer found our paper “... solid and easy to understand ....”, “very competitive against SOTA”, and  “the writing is simply excellent”. We provide the response to each concern in the following.
>
> **1. Theoretical justification on local PCA-SDF over local implicit grid-based methods.**
>
> i) We agree that it would be nice to have a theoretical justification on why local PCA-SDF outperforms the local implicit grid-based methods. Yet at this moment it appears not straightforward to do that. On the other hand, our idea in itself is well motivated. During the implicit network training, the local implicit grid-based methods (*i.e.,*  DeepLS, [60]) treats each point in the voxel as an independent sample to infer its occupancy.   In contrast, there is a strong prior on how the SDFs of neighboring points are correlated and vary across all voxels. For example, SDF of a voxel on the table surface is a planar surface. Our local PCA-SDF is able to leverage this prior and predict the SDFs of all points within a voxel jointly instead of individually, leading to more robust and accurate reconstruction.
>
> ii) NGLOD and ACORN further introduce hierarchical feature grids strategies into the local implicit network based on octree or quadtree representation. Although achieving high accuracy, the *current versions* of these two methods do not offer solutions for learning the mapping from image features to the 3D features grids, which limits their application for single image reconstruction.
>
>
> **2. Ablation study on Positional Encoding operator.**
>
> Please refer to **the response of Reviewer 2Xpx Q4.**
>
>
> **3. Angle representation.**
>
> In our experiments, we restrict the range of the angles to [-π,π]. During training, our model allows the angles to exceed the range [-π,π]. However, we simply apply the clamp operator to the predicted angles into the range [-π,π] during inference.
>
>
>  **4. Reconstruction at scene level or box domain.**
>
> Yes, our framework performs reconstructions at the scene level. As the reasons we mention in  **the response of Reviewer NjXM  Q3 --  i),** box-based single object reconstruction methods are not desirable for reconstructing multiple objects within a scene. It's really a good suggestion that we should conduct ablation studies against one which performs reconstruction within each object’s bounding box domain.  Unfortunately, we are unable to finish up training the model due to the tight schedule. We thank the reviewer for bringing this to our attention and we will add this comparison in the revision.
>
> **5. The Chicken-and-egg problem between grid position and 3D detection.**
>
> We agree that the grid position estimation and 3D detection are somehow chicken-and-egg problems. Consequently, we assume the camera calibration matrix is known, and then we can empirically pre-define the 3D grid, *e.g.,* we use a voxel grid of size 6.4m×6.4m×6.4m for the indoor ScanNet-MDR dataset (L262-263). We did acknowledge this assumption in L42, L327-329.
>
>
>  **6. NMS for the heatmap.**
>
> According to CenterNet [26], we do NOT need IoU-based NMS or other post-processing. The peak keypoint extraction serves as a sufficient NMS alternative and can be implemented by using a 3×3×3 max-pooling operation in our experiment.
>
>
>  **7. Comparison to MeshRCNN on Pix3D.**
>
> As suggested, we report the comparison to MeshRCNN on the S1 split of the Pix3D dataset. Our method achieves better than MeshRCNN in both Chamfer Distance-L2 and IoU metrics.
>
> |                                       |   Mesh R-CNN   |         Ours        |
> |---------------------------------|:---------------------:|:-------------------:|
> |    Chamfer Distance-L2  |         0.0111       |        0.0082      |
> |     IoU (128-dim voxel)   |         28.3%        |        42.6%       |
>
>
> **8. Feature smearing issues.**
>
> As claimed in L43-44, L136-140,  we believe that the positional encoding operator could mitigate the feature smearing issue, which enables the reconstruction module to hallucinate invisible back-facing object parts and truncated parts. Also please refer to the visualizations of the learned voxel features before and after the positional encoding operator in Fig. 2 of the Supp. It can be observed that after the positional encoding, the boundary of the voxel features, in both the bird’s eye view and side view, become more sharp compared to the ones before the positional encoding. Since feature smearing often produces in-between depth pixels and blurred boundary, we view the improved sharpness of voxel features as an indication of reduced feature smearing.
>
>
> **9. Technical details.**
>
> i) *Homogeneous coordinates.* Thank you for pointing out this issue. Actually, Eqn. 1 is to construct the 8 corners of the 3D bounding box and it should be rewritten as
> $R(\alpha,\beta,\gamma) \begin{bmatrix}
> l & l & -l & -l & l & l & -l & -l \\\\
> -h & -h & -h & -h & h & h & h & h \\\\
> w & -w & -w & w & w & -w & -w & w \\\\
> \end{bmatrix}/2 + repeat(\mathbf{c}_{3d},1,8)$
>
>  We do not need to project the 3D bounding boxes into 2D spaces. So they do not need to be in homogeneous coordinates.
>
> ii) *Interpolation in L199.* We just want to mention that, our PCA-SDF representation can infer 3D shape at any resolution as other local implicit grid methods (*i.e.,*  NSVF and NGLOD) with a simple interpolation operator.
>
> iii) *L2 loss in Eqn.3.* Thank you for your insight. L2 loss is more consistent between modeling and learning. As suggested, following the same experimental setting in Sec. 4.5, we train a model by using L2 loss in Eqn. 3 and the reconstruction accuracy is 40.1% (IoU) -- slightly better than L1 loss.
>
> iv) *L213.* Thank you for pointing this out. Here we actually utilize the mean absolute error loss on the 3D bounding box regression.
>
>
> **10. Further comments.**
>
> Thank you for your valuable comments to improve the original manuscript. We have carefully and thoroughly revised the manuscript in terms of captions, terminologies, and symbol denotations.

---

### Official Review · Reviewer_NjXM · 2021-07-15

**Rating:** 6
**Confidence:** 4

**Summary:**

This paper presents a joint framework for detecting multiple 3D objects from input images while also reconstructing the 3D shapes. The proposed method improves upon previous work [20,21] for joint 3D object detection and reconstruction mostly at the system/architectural level. The authors also propose a new way of compactly representing 3D shapes with signed distance functions (SDF) using PCA that can be naturally incorporated into the framework. Experiments are benchmarked with synthetic multi-object ShapeNet renderings, Pix3D, and a new dataset with annotations for multi-object detection & reconstruction.

**Limitations And Societal Impact:**

Adequately addressed

**Main Review:**

Strengths:
+ A nice overview of related literature is provided.
+ The method description is clear, and it is easy to follow and understand. The overview diagram is also helpful in understanding the entire framework.
+ The proposed PCA-SDF representation is particularly interesting, as high-quality SDF shapes can be broken down into a collection of SDF parts that can be compactly represented using a pretrained dictionary, which is generalizable to out-of-category shapes and greatly speeds up inference time.
+ Experimental results show that the proposed system outperforms the baseline methods on different benchmarks.

Weaknesses:
- The introduction doesn't give much context on where this work currently stands in the problem scope of 3D object detection and reconstruction. Why is joint 3D detection/reconstruction important? It is also unclear what the current major problems are for this specific task, and the intro is written rather much like another related work section.
- In the related work section, it is unclear how two-stage pipelines [17,19,51] would be inferior and why one would not prefer these methods over the proposed (L87-88). Personally, I do not see how decoupling detection and object reconstruction would be undesirable. It is also unjustified why depth estimation in [21] is undesirable and why the proposed CenterNet-3D can improve detection accuracy (L92-93,150-151). The authors should also discuss the difference between this work and [A], a very related prior work on voxel-based 3D detection/reconstruction.
- "CenterNet" seems to be an important part of the pipeline (detection) and the term has been repeated throughout the paper, but it was never explained what CenterNet is. I think the authors should devote at least a paragraph explaining what CenterNet is and how it is useful for the design of the proposed method. The authors should not assume it is a commonly known concept.
- It is unclear which component in the system is the most crucial to the performance improvement. Is it the use of voxel features, the design of CenterNet-3D, or the use of the new PCA-SDF representation? It is also unclear from the experiments section, as the authors reported results without justifying why there was the improvement. In the current form, I cannot see what the key message or takeaway is from this paper.
- It is difficult to compare the presented visual results. For one, the ground-truth 3D CAD models are not shown (Fig. 4, 5, 6) so it's hard to tell which method is actually better, despite the results from the proposed method looking slightly more plausible. Ground-truth shapes were shown in the supplementary videos but lacks baseline comparisons, and I think having all of them together is important for evaluation. On a side note, the visualization between CoReNet and the proposed in Fig. 4 are different, which slightly prevents a direct comparison.
- I appreciate the authors' effort to create a new real-world dataset for more supportive evaluation, but I think there should be at least some visualization of what the new dataset looks like by including some example images and ground-truth CAD models processed from Scan2CAD. The quality of the dataset, and thus the evaluation quality, is currently unclear. Also, why is [21] not compared against on this new dataset? Evaluating the detection mAP but only comparing against [20] (which does not perform detection) is not meaningful. Finally, will the new dataset be released to the public?

I think in general, the authors seem to aim to get better benchmarking results as the ultimate goal, but it is not clear what components really contributes to the system. As many of the components seem orthogonal (e.g. 3D feature lifting, coarse-to-fine voxelization, and PCA-SDF), it would be good to analyze and ablate each individual components in depth, and show that an integrated system with all the improvements would be most desirable.

Other minor problems:
- Why are $\delta_h$, $\delta_w$, $\delta_l$ residual sizes? This term is misleading as they are actually the logarithm of the sizes.
- Concatenating (x,y,z) coordinates in convolutional operations is not positional encoding, but rather CoordConv [B]. Positional encoding refers to a deterministic mapping for coordinate-based representations, which is different. On a side note, the original concept for positional encoding should be from Transformers [C] instead of NeRF [66].
- What does "might receive the same 2D feature" mean in L136?
- What do "200 surfaces" mean (L231,264)? What counts as "a surface"?
- The citation for focal loss is wrong -- it should be [D].
- It is unclear why predicting a higher-dimensional code is more challenging for neural networks (L290).

[A] Tung et al. "Learning Spatial Common Sense with Geometry-Aware Recurrent Networks." CVPR 2019
[B] Liu et al. "An Intriguing Failing of Convolutional Neural Networks and the CoordConv Solution." NeurIPS 2018
[C] Vaswani et al. "Attention Is All You Need." NIPS 2017
[D] Lin et al. "Focal Loss for Dense Object Detection." ICCV 2017

**Time Spent Reviewing:**

6hr

---

> ### Author Response · Authors · 2021-08-10
> **Response to Reviewer NjXM**
>
> We highly appreciate the time and effort the reviewer has invested in reviewing our paper. The reviewer recognized the strengths and novelties of our method in terms of clear method description, interesting and efficient PCA-SDF, and superior performances. We address the concerns raised by the reviewer in the following.
>
> **1. Ablation studies on each component.**
>
> i) In Table 2, we show the detailed ablation studies: joint framework vs. separate modules, CenterNet-3D vs. conventional CenterNet, and local PCA-SDF vs. DeepLS. The results help us to identify both the **source**and **amount**of improvement. For example, 37.6% ----> 38.2% (IoU) is a reconstruction improvement due to the conventional CenterNet module while  37.6% ----> 39.8% (IoU) is due to the proposed CenterNet-3D module. For detection improvement, 24.9% ----> 26.7% (mAP@0.15) is due to the proposed local PCA-SDF, and  23.2% ----> 26.7% (mAP@0.15) is due to the proposed CenterNet-3D module.
>
> ii) We additionally ablate the effect of the positional encoding operator and please refer to **the response of Reviewer 2Xpx Q4.**
>
> iii) Based on those results, we believe that, in the proposed joint framework, the two tasks are able to boost each other.
> 3D voxel features can improve 3D detection by a large margin ( 15.1% relative improvements).
> The positional encoding operator does mitigate the feature smearing issue and further improve both detection and reconstruction performances.
> We will revise and add more theoretical and experimental analysis in the revision.
>
>
> **2. The revision of the Introduction section.**
>
> Thank you for the valuable comments to improve the introduction section. We have carefully and thoroughly improved this section.
>
>
> **3. Comparison to pipelines [17,19,51,21].**
>
> i) [17,19,51] first detect objects in the 2D image. Then, for each individual 2D RoI, they independently infer its 3D shapes with single-object reconstruction modules. Instead of producing a globally coherent reconstruction for multiple objects of the image, these methods suffer from inferring object-to-object interactions (e.g., occlusion). For example, for the image example in Fig. 2 (main paper), the 2D Rols of the chair and table share largely overlapping, on which the box-based single object reconstruction methods would fail. In addition, independently inferring 3D shapes cannot guarantee that reconstructed objects should not intersect or collide with each other.  We believe there is a minor misunderstanding and we will carefully revise L87-88.
>
> ii) Unlike CenterNet in [21], CenterNet-3D avoids directly estimating depth values from image features. Moreover, our voxel features have a Euclidean structure and are aligned with the world coordinate, which provides additional depth constraint in each voxel, leading to reliable depth dues for the CenterNet-3D module.
>
>
> **4. Difference to Tung et al. (CVPR’ 19).**
>
> Thank you for bringing up this nice work. This is certainly a relevant work and we will cite and discuss it in the revision. We see the following differences between this work and ours.
>
> i)  The goal is different. Tung *et al.* perform 3D detection and segmentation, while our method focuses on 3D detection and fine-grained reconstruction.
>
> ii) The input is different. Tung *et al.* require a sequence of frames as input, while our method only takes **a single image**as input.
>
> iii) The approach is different. Tung *et al.* utilize a 3D maskRCNN for 3D detection and segmentation, while we propose novel CenterNet-3D and local PCA-SDF to better tackle the tasks of 3D detection and reconstruction.
>
>
> **5. Introduction of conventional CenterNet.**
>
> This is a good suggestion. We will add one paragraph to introduce the conventional CenterNet method.
>
>
> **6. Better to show visual results.**
>
> We agree that the ground-truth shapes should really be added in Fig. 4,5,6 and we will add them. For Fig. 4, we will re-do it by using the same rendering setting both for CoReNet and our method.
>
>
> **7. Comparison of the new dataset.**
>
> As suggested, we train Points2Objects on the ScanNet-MDR dataset for comparison. Following the same experimental setting of the original paper, we select k=50 shapes of each class by K-means to form the shape pool. The evaluated detection and reconstruction accuracies are 23.6(mAP@0.15) and 31.9(IoU) respectively, which is worse than ours (26.7(mAP@0.15)  and 39.8(IoU). Since the retrieval-based methods severely rely on the size and diversity of the shape pool, Points2Objects performs substantially worse 3D reconstruction than ours. We will make the dataset publicly available.
>
>
> **8. Minor issues.**
>
> i) Yes, \\( \delta_h, \delta_w, \delta_l  \\) is not the residual of object size, we will fix the description.
>
> ii) We will cite the appropriate references for positional encoding and focal loss.
>
> iii) (L136) During the 2D-to-3D lifting process, multiple voxels along the projection ray might be assigned to the same image feature.
>
> iv) (L231,264) “200 surfaces” means 200 shapes.
>
> v) (L290) It’s more challenging for the network to learn the mapping from 2D image features to higher-dim PCA coefficient codes.
> Lower-dim PCA coefficient codes often make training more efficient and can help make more accurate feature mapping.

---

> > ### Comment · Reviewer_NjXM · 2021-08-24
> > **Thanks for the response**
> >
> > Thanks for the detailed response. I think most of my concerns have been adequately addressed, so I'm happy to update my rating to a borderline accept.

---

### Official Review · Reviewer_2Xpx · 2021-07-16

**Rating:** 8
**Confidence:** 4

**Summary:**

The paper proposes a novel method for 3D detection and reconstruction of multiple objects from single images, that combines 2D feature extraction and 2D-3D lifting with a modified (3D) version of the CentreNet detector and  novel coarse-to-fine reconstruction stage.

**Limitations And Societal Impact:**

-

**Main Review:**

Positive:
* The paper is well organised, very well written and easy to understand.
* The approach has several novel components and (i) appears to produce state of the art results while (ii) improving on inference speed over other competing methods.
* The experimental section is quite extensive and does appear to cover most essential benchmarks and ablations.

Negative / Questions:
* Proposed 1-4 is somewhat confusing — maybe put more text in the caption. I’d be good to expand the text on lines 275 - 282.
* What if you had bigger rotation perturbations? At what angled would we expect PCA-SDF to fail?
* Are there any advantages of PCA-SDF over using PCA over a standard occupancy?
* Is the importance of the positional encoding benchmarked anywhere?
* Did you do any ablation on the size of the PCA-SDF local volume? Would smaller volumes require more fewer basis functions and generalize better, or does the size of the volume not matter much?
* Some typos (eg line 283).

Overall, I am very much in favour of acceptance. The proposed approach is principled and produces very good results, and the paper is very well written.

**Time Spent Reviewing:**

2

---

> ### Author Response · Authors · 2021-08-10
> **Response to Reviewer 2Xpx**
>
> We thank the reviewer for the constructive and valuable feedback. The reviews are positive about the method’s novelty as well as compelling results. For instance, the reviewer wrote “The approach has several novel components ...”, “... produce a state of the art results ...” and “... the paper is well organized ...”. We answer the reviewer's questions and comments below.
>
> **1. Ablation studies in Table 2.**
>
> i) The proposed framework includes two novel modules: CenterNet-3D detection and local PCA-SDF-based reconstruction. In this experiment, we first ablate the performances of individual modules by removing one of the modules. Based on the results in Table 2 (Proposed-1, Proposed-2 and Proposed), we can conclude that our joint framework performs better than solving either task exclusively.
>
> ii) To further evaluate the effectiveness of the proposed CenterNet-3D detector and local PCA-SDF shape representation. We train models by replacing one of the modules with the conventional CenterNet detector or DeepLS shape representation respectively. The comparisons show the respective contribution of each of our two novel modules.
>
> iii) For more analysis, please refer to **the response of Reviewer NjXM Q1.** Thank you for pointing out this issue, we will add more descriptions on this in the revision.
>
>
> **2. Local  PCA-SDF on large rotation perturbations.**
>
> As suggested, we conduct an experiment to investigate whether our local PCA-SDF is sensitive to **large**rotation perturbations. Following the same setting in Fig.5(c),  we apply rotation variations (±90°, ±70°, ±50°, ±45°, ±30°, ±10°) on one random axis (x, y, or z) to testing surfaces of the 5 categories. The Chamfer Distance-L2 (CD, multiplied by $10^3$) are reported together with results on small rotation perturbations (Fig. 5(c)) as follows.
>
> |              |   ±90°    |   ±70°   |    ±50    |    ±45°   |    ±30°   |    ±10°   |   *±4°* |   *±2°*  |
> |------------|:----------:|:----- ----:|:----------:|:----------:|:----------:|:-----------:|:--------:|----------:|
> |     CD    |  0.0251 | 0.0272  | 0.0296  |  0.0306 |  0.0281 | 0.0256   |0.0210 | 0.0229 |
>
> As can be observed, the errors are small across all rotation angles. The potential reason that attributes to the largest error in ±45° is that the ±45° rotations may lead to a much different boundary voxel distribution with local PCA-SDF training voxels.
>
>
> **3. PCA-SDF vs. PCA-Occupancy.**
>
> Generally, PCA is more suitable to be applied to continuous-valued SDF than binary-valued occupancy. Moreover, SDF is more straightforward to utilize a simple strategy (*i.e.,* average pooling on the boundary SDF values from neighboring voxels) to avoid inconsistent surface estimates at the voxel boundaries. Nonetheless, following the same setting in Table 4, we report the average Chamfer Distance-L2 (CD, multiplied by $10^3$) of PCA-Occupancy (PCA-O).
>
> |              | PCA-O ($l_B$=32)|PCA-O ($l_B$=64)|PCA-O ($l_B$=125)|PCA-SDF ($l_B$=125)|
> |------------|:------------------------:|:-----------------------:|:-------------------------:|:-------------------------:|
> |     CD    |          2.354            |           0.761         |          0.124             |          0.025             |
>
> As can be observed, PCA-Occupancy is much worse than our PCA-SDF (please also refer to Table 4).
>
>
> **4. Ablation study on the Positional Encoding operator.**
>
> We show the visualizations of the learned voxel features before and after the positional encoding operator in Fig. 2 (supplementary). It can be observed that the positional encoding indeed enhance the voxel features. For instance, the boundary of the voxel features, in both the bird’s eye view and side view, become more sharp compared to the ones before the positional encoding. As suggested, we also retrain a model without the positional encoding operator on the ScanNet-MDR dataset. The detection and reconstruction accuracies are 24.4(mAP@0.15) and 34.2(IoU) respectively, which is worse than ours (26.7(mAP@0.15)  and 39.8(IoU)) in Tab.2). We thank the reviewers for bringing this to our attention and we will add this ablation study in the main paper.
>
>
> **5. Ablation study on the size of local volume.**
>
> We did report this ablation study in Fig. 5(b) and L283-290.

---

> > ### Comment · Reviewer_2Xpx · 2021-08-27
> > **Maintaining rating**
> >
> > Thank you for the comments. I am happy to maintain my rating.

---

### Official Review · Reviewer_7tQr · 2021-07-17

**Rating:** 6
**Confidence:** 3

**Summary:**

The authors propose a method for 3d detection and reconstruction from a single image. The main novelty of the paper in my opinion is the PCA_SDF shape representation which improves inference time and helps capture details in the reconstruction.

**Limitations And Societal Impact:**

The authors do not discuss this. Perhaps, a sentence or two can be added beyond stating that there are no obvious negative impacts. The non-obvious ones are the ones to think and discuss.

**Main Review:**

Strong Points
- The paper improves the SOTA in 3d reconstruction from a single image
- The PCA_SDF improves details while being computationally effiicient.

Weak Points
- The paper does 3d detection using a 3d centernet. It is unclear what is the objective of performing this detection. There is a large body of work on monocular 3d object detection that this paper completely ignores in related work and in experiments. Is the 3d detection a way to regularize the network? Does the network benefit from multi-task effects?
- The overall novelty of the approach is marginal. Each module is isolation has been well studied. In that case, the main novelty is piecing these individual elements together which is more engineering that research. The fact that the authors do both detection and reconstruction dilutes the contributions of the paper. The authors should focus on reconstruction and dive into details.
- The PCA_SDF borrows ideas from EIgenSDF [63] and Jiang et. al. [60]. The authors need to do more ablation studies by swapping out their PCA-SDF with eigen-sdf/ local implicit grids and be more thorough about the benefit of the pca-sdf. A lot of work on sdf has emerged since deepsdf and it is important to validate this work against the latest and greatest.

Post Rebuttal: Based on the response of the authors, and having evaluated the other reviews, I am updating my rating for this paper. Although some of the other reviewers mirrored my concerns of the paper, I agree that the improvement in results with simple innovations is of broad appeal to the research community working on 3D reconstruction.

**Time Spent Reviewing:**

1.5

---

> ### Author Response · Authors · 2021-08-10
> **Response to Reviewer 7tQr**
>
> We thank the reviewer for the insightful reviews and for acknowledging the strong points of our reconstruction module, efficiency, and accuracy. In the following, we address the reviewers’ concerns.
>
> **1. The objective of performing 3D detection.**
>
> i) We believe that simultaneously inferring both tasks of 3D shapes and 3D bounding boxes of multiple objects from a single image is more beneficial than either task. Our coherent 3D reconstructions manifest spatial occupancy that could help 3D object detection. Meanwhile, 3D detection enables us to locate the 3D shapes in the physical world with an absolution scale. Also, 3D detection provides class priors that could benefit 3D reconstruction.
>
> ii) We also provide evidence that the proposed joint framework does perform better than solving either task exclusively (Please refer to L272-274 and Table 2 Proposed-1 vs. Proposed-2 vs. Proposed). Thanks for pointing this out, we will highlight this conclusion early in Section 1.
>
> iii) We will add an additional paragraph to review the monocular 3D object detection methods in the related work part. In the experiment, we actually compare with the conventional CenterNet on detection in Table 2 (@mAP: 23.2 vs. 26.7). Nonetheless, we will add comparisons to more baselines in the revision as suggested.
>
>
> **2. Claims adjustment.**
>
> i) In this work, we propose a voxel feature representation for **joint**3D detection and reconstruction of **multiple objects**from a single image. Both tasks can benefit from each other through the proposed joint framework.
>
> ii) We would like to emphasize that the proposed CenterNet-3D detection module and local PCA-SDF-based coarse-to-fine 3D reconstruction module are **novel**and have not been studied in previous literature. We provide evidence that CenterNet-3D detection improves detection performance over conventional CenterNet (Tab. 2 Proposed-3 vs. Proposed). Meanwhile, local PCA-SDF shape representation provides more accurate reconstructions and **orders of magnitude**faster inference than baseline (Tab. 4 DeepLS ($l_B=125$) vs. PCA-SDF ($l_B=125$)).
>
> iii) Thank you for the suggestion, we will add more details about the 3D reconstruction in the revision.
>
>
> **3. Comparison to EigenSDF/local implicit grids-methods.**
>
> i) We did compare with the latest local implicit grids-based method, DeepLS [28] (ECCV’ 20) in Tab.2 (Proposed-4) and Tab.4. As can be observed, our local PCA-SDF achieves a lower reconstruction error and 100× faster inference efficiency than DeepLS. We did not compare with Jiang et al. [60] (CVPR’ 20) because 1) the local implicit grid representation in [60] is similar to DeepLS, 2) DeepLS is a more recent work than [60], and 3) [60] does not release the training code.
>
> ii) As suggested, we conduct additional experiments to compare our Local PCA-SDF representation with EigenSDF [63] and more DeepSDF-based methods, *i.e.,*  IF-Net (CVPR’ 20), Convolutional occupancy network (CON, ECCV’ 20) and PatchNet (ECCV’ 20). For EigenSDF, we construct the global PCA model on training samples with a resolution of 128×128×128. The dimension of latent codes is set to 256 as DeepSDF. IF-Net, CON, and PatchNet are extended works of implicit shape representation which can represent high-quality 3D shapes. Following the same training/testing split in Table 4 (main paper), we train their models based on the publicly released codes of the respective authors. In contrast to us, during testing, these three methods require voxels or points as inputs to obtain local features for implicit networks. We report the average Chamfer Distance-L2 (CD) (multiplied by $10^3$) over 5 categories as follows. As can be observed, our local PCA-SDF achieves the best performance in shape representation.
>
> |              |    EigenSDF    |    IF-Net    |     CON    |     PatchNet    |     Ours    |
> |------------|:-------------------:|:-------------:|:-------------:|:------------------:|:------------:|
> |     CD    |        1.569          |    0.058      |     0.069     |          0.051      |    0.028     |

---

### Decision · Program_Chairs · 2021-09-27

**Decision:**

Accept (Poster)

**Comment:**

This paper initially had mixed reviews (8,8,5,5). After reading the rebuttal and checking the other reviewers' comments, the two negative reviewers were convinced and raised their score to 6. Reviewer 7tQr appreciated the simple innovation this paper offers, which is of broad appeal to the 3D reconstruction community. Reviewer NjXM stated that the rebuttal addressed their concerns adequately. Hence, this paper now has 4 positive scores and it should be accepted.